# Quantitative Volumetric Enamel Loss after Orthodontic Debracketing/Debonding and Clean-Up Procedures: A Systematic Review

**Gaetano Paolone [1],*** , **Mauro Mandurino [1]** , **Sofia Baldani [1]** , **Maria Giacinta Paolone [2]** , **Cecilia Goracci [3]** , **Salvatore Scolavino [4]** , **Enrico Gherlone [1]** , **Giuseppe Cantatore [1] and Giorgio Gastaldi [1]**

[1] Dental School, IRCCS San Raffaele Hospital, Vita-Salute University, 20132 Milan, Italy
[2] Independent Researcher, 00100 Rome, Italy
[3] Department of Medical Biotechnologies, University of Siena, 53100 Siena, Italy
[4] Independent Researcher, 80035 Nola, Italy
* Correspondence: paolone.gaetano@hsr.it; Tel.: +39-02-2643-2970

**Abstract:** Objectives: To conduct a systematic review assessing quantitative enamel loss occurring after orthodontic debonding and clean-up procedures. Materials and Methods: A systematic search following the Preferred Reporting Items for Systematic Reviews and Meta-Analyses (PRISMA) statement was performed on different databases (Embase, Medline, Scopus, Web of Science) for papers investigating volumetric enamel loss due to bracket and clear aligner attachment debonding and/or clean-up procedures. Studies investigating in vivo and in vitro articles published in the English language until 16 July 2022 were included. The study selection was then performed by two authors who screened the abstracts independently. Results: Of 421 screened abstracts, 41 articles were selected for full-text analysis. Finally, nine studies were included in this review. No in vivo papers were retrieved. In vitro papers investigated volumetric loss caused by the removal of metal brackets (n = 7), ceramic brackets (n = 1), and both (n = 1). The clean-up procedure varied among all investigations. Impressions at baseline and after debonding/clean-up were superimposed, and the volumes were subtracted using different 3D digital analysis software. Among all included studies, the volumetric loss of enamel ranged from $0.02 \pm 0.01$ mm$^3$ to $0.61 \pm 0.51$ mm$^3$ per tooth. Conclusions: Debonding and clean-up procedures produce enamel loss. The debonding/clean-up procedure that is able to cause the least enamel volume loss has yet to be identified.

**Keywords:** debonding; clean-up; enamel loss; debracketing; volumetric loss; enamel damage

## 1. Introduction

In orthodontic treatments performed with fixed appliances or clear aligners, brackets or attachments are bonded to enamel [1,2]. Despite a high bond strength between adhesive and enamel being sought to counteract forces due to orthodontic therapy, it increases the possibility of enamel damaging during debonding and clean-up procedures [3,4]. At the end of the orthodontic treatment, a major concern is the removal of those devices and adhesive resin from enamel surface with less damage to the enamel surface as possible [1,5]. Enamel damage may be a consequence of enamel cracking during debracketing procedure or grinding residual adhesive during the clean-up procedures. The outermost layer of enamel should remain as intact as possible since it is characterized by higher mineral content and fluoride than the deeper layers [6]. The loss of superficial enamel might cause a decrease in the resistance of enamel to the organic acids available in the oral environment, making it more prone to demineralization.

Debonding and clean-up procedures are time consuming, cumbersome, and cause mental stress on both patients and orthodontists [5,7–9]. Improper tools and devices, together with the haste of finishing the treatment, may lead to enamel damage. Furthermore,

orthodontic adhesives used for bracket bonding are difficult to distinguish from the surrounding tooth structure, making them very difficult to remove without damaging the tooth structure. Hence, orthodontists, during adhesive removal, may accidentally affect excessive amounts of healthy tooth structure that may need avoidable future restorative procedures [10,11].

Clean-up methods after bracket debonding comprise hand or rotary tools such as diamond burs [12], tungsten carbide burs [13–15], silicone points [16], fiber-reinforced composite burs [17], and aluminum-oxide-coated disks [14,18,19]. Nevertheless, currently, there is no consensus on which technique must be preferred for orthodontic clean-up [3,20].

Qualitative analysis of adhesive residuals left after clean-up is generally performed through the ARI index [21] (0 = no adhesive; 1 = less than half of the; 2 = more than half of the adhesive; and 3 = all adhesive left with distinct bracket impression). Enamel damage assessment has also been performed mainly through qualitative methods such as the Enamel Damage Index (EDI) [22] (0 = smooth surface no scratches and perikymata still visible; 1 = only a few superficial scratches; 2 = several deeper grooves and scratches; 3 = grooves and scratches are detected with the naked eye) but seldom through quantitative analysis.

So far, in fact, although new technologies such as micro-computed tomography (micro-CT) systems [23], profile projectors [24], null-point contact stylus systems [25], planer surfometers [26,27], laser scanning [5,5], 3D contact profilometry [28], and intra-oral scanners [29] could allow the quantitative evaluation of the enamel loss after debonding and clean-up procedures, only few papers have directly investigated it. Therefore, to investigate the quantitative enamel loss during debonding as well as during clean-up procedures, a systematic literature review was conducted.

## 2. Materials and Methods

### 2.1. Protocol and Registration

The protocol for this systematic review was constructed a priori based on Cochrane Handbook for Systematic Reviews of Interventions 5.1.0. The protocol was registered online at https://osf.io with DOI 10.17605/OSF.IO/SNYEC. This systematic review was written according to the Preferred Reporting Items for Systematic Reviews and Meta-Analyses (PRISMA) statement [30]. The focused question was formulated according to the PICO framework to develop the search strategy:

Population (P): buccal or lingual surfaces with orthodontic brackets or clear aligner attachments.

Intervention (I): debonding/clean-up procedures.

Comparison (C): sound surfaces before bracket or attachment application.

Outcome (O): enamel loss.

### 2.2. Eligibility Criteria

2.2.1. Inclusion Criteria

- Studies written in the English language;
- In vivo/in vitro studies that investigated quantitative volumetric analysis of enamel loss;
- Studies that considered bracket and attachments debonding and/or clean-up.

2.2.2. Exclusion Criteria

- Type of study: case report, technical report, reviews;
- Studies evaluating splinting.

### 2.3. Information Sources and Search Strategy

Two reviewers (G.P and M.M.) conducted a search for English-language articles published in dental journals until 16 July 2022. Electronic searches were conducted on the following different databases: MEDLINE/PubMed, Embase, Scopus, and Web of Science. The search strategy was designed to find in vivo or in vitro articles that evaluated quantitative volumetric enamel loss after debonding and/or clean-up procedures. All search strategies

relied on the search strategy developed for PubMed (Table 1) and appropriately adjusted to each database to account for differences in controlled vocabulary and syntax rules.

**Table 1.** Search conducted in Medline/PubMed database.

| Search | Query |
|---|---|
| #1 | ("debonding") OR ("cleanup") OR ("clean-up") OR ("debracketing") OR ("adhesive removal") OR ("cement removal") OR ("composite removal") OR ("bracket removal") OR ("adhesive clearance") OR ("composite clearance") |
| #2 | ("damage") OR ("defect") OR ("crack") OR ("loss") OR ("micro-crack") |
| #3 | #1 AND #2 AND ("enamel") |

For the selection of studies, two authors (M.M. and S.B.) independently reviewed titles and abstracts of the studies according to the inclusion criteria. Final inclusion of studies was based on screening and assessing full texts and with consensus of the authors of the current review. The Rayyan website was used to automate duplicate removal and facilitate inclusion decisions. The reference lists of all included papers were finally checked for any potential article loss.

*2.4. Data Collection and Synthesis Methods*

A standardized data extraction form was made using Excel software (Microsoft Corporation, Redmond, WA, USA), collecting the following data (Table 2): author, publication year, type of brackets, adhesive system, debonding procedures, clean-up procedures, 3D surface acquisition device, acquisition time, type of model, digital analysis software, type of specimens, and minimum and maximum mean volumetric loss ($mm^3$) per tooth. During data extraction, all information was found in the articles texts, and reviewers did not need to contact the authors.

**Table 2.** Risk of bias.

| | Clearly Stated Aims/Objectives | Detailed Explanation of Sample Size Calculation | Detailed Explanation of Sampling Technique | Details of Comparison Group | Detailed Explanation of Methodology | Operator Details | Randomization | Method of Measurement of Outcome | Outcome Assessor Details | Blinding | Statistical Analysis | Presentation of Results | Total Score | Final Score % | Risk of Bias |
|---|---|---|---|---|---|---|---|---|---|---|---|---|---|---|---|
| Tufekci et al. 2004 [25] | 2 | 0 | 2 | 2 | 2 | 0 | 0 | 2 | 0 | 0 | 1 | 2 | 13 | 54.17 | MEDIUM |
| Banerjee et al. 2008 [28] | 2 | 0 | 2 | 2 | 2 | 0 | 0 | 2 | 0 | 0 | 2 | 2 | 14 | 58.33 | MEDIUM |
| Ryf et al. 2012 [5] | 2 | 0 | 1 | 1 | 2 | 0 | 1 | 2 | 0 | 0 | 1 | 1 | 11 | 45.83 | HIGH |
| Janiszewska-Olszowska et al. 2014 [31] | 2 | 0 | 2 | 1 | 2 | 0 | 0 | 2 | o | 0 | 2 | 2 | 13 | 54.17 | MEDIUM |
| Suliman et al. 2015 [29] | 2 | 0 | 2 | 1 | 2 | 0 | 0 | 2 | 0 | 0 | 1 | 2 | 12 | 50 | MEDIUM |
| Janiszewska-Olszowska et al. 2015 [32] | 2 | 0 | 2 | 1 | 2 | 0 | 0 | 2 | 0 | 0 | 2 | 2 | 13 | 54.17 | MEDIUM |
| Stadler et al. 2019 [33] | 2 | 0 | 2 | 2 | 2 | 0 | 1 | 2 | 0 | 0 | 2 | 2 | 15 | 62.5 | MEDIUM |
| Cesur et al. 2022 [34] | 2 | 0 | 2 | 2 | 2 | 0 | 0 | 2 | 0 | 0 | 2 | 2 | 14 | 58.33 | MEDIUM |
| Engeler et al. 2022 [35] | 2 | 1 | 2 | 2 | 2 | 1 | 0 | 2 | 0 | 1 | 2 | 2 | 17 | 70.83 | LOW |

*2.5. Risk of Bias*

Risk-of-bias assessment followed the QUIN tool [36] (risk-of-bias tool for assessing in vitro studies conducted in dentistry) and was performed manually by two reviewers (G.P. and M.M.). Its domains and relative scores are presented in Table 2.

**3. Results**

*Study Selection and Study Characteristics*

The study selection process according to the PRISMA checklist is reported in Figure 1.

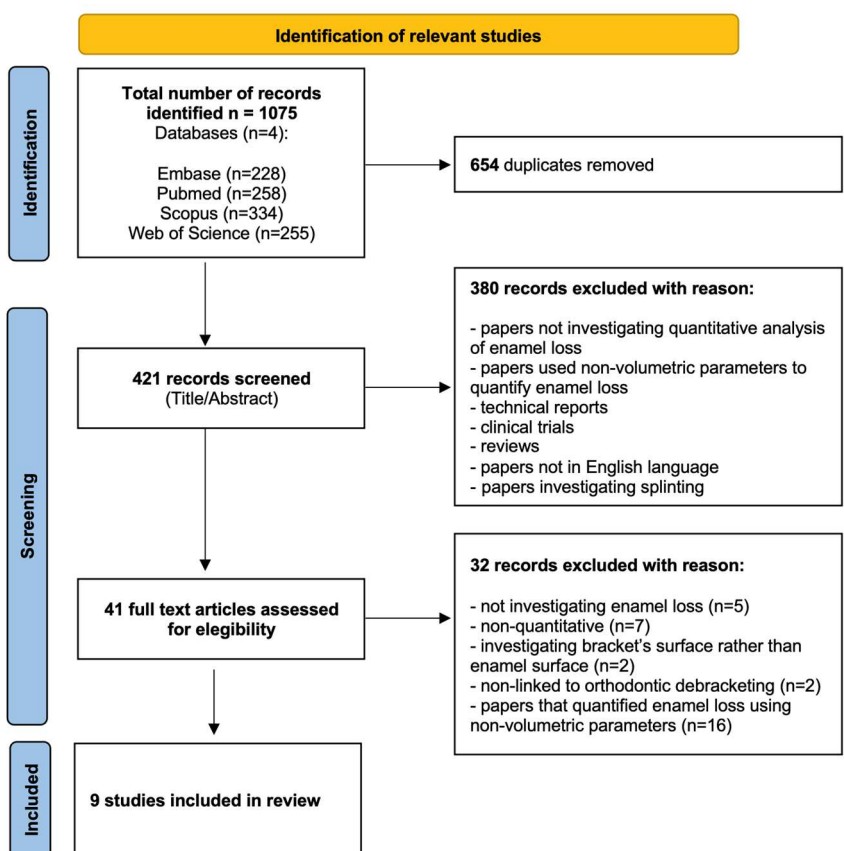

**Figure 1.** Identification of relevant studies.

One thousand seventy-five records were identified through database searching. Following removal of duplicates, 421 records were screened by title and abstract. During the screening process, in agreement with all the reviewers, 380 records were excluded as not relevant to the subject, and 41 articles were selected for full-text analysis; 32 studies were excluded since they did not meet the eligibility criteria due to the following reasons: they did not investigate enamel loss (5 studies) or did not perform quantitative evaluation (7 studies), or they were studies considering the bracket surface rather than the enamel surface (2 studies), studies not linked to orthodontic debracketing (2 studies), or articles that quantified enamel loss using non-volumetric parameters (16 studies). Finally, nine studies [5,25,28,29,31–35] were selected for the systematic review. Risk-of-bias assessment for the included studies is shown in Table 2.

Results of Individual Studies and Synthesis of Results

Characteristics of the nine studies included in the systematic review are presented in Tables 3 and 4.

**Table 3.** Included studies and assessed variables, part 1.

| | Type of Bracket | Adhesive System | Debonding Procedure | Clean-Up Procedure | 3D Surface Acquisition Device | Acquisition Time |
|---|---|---|---|---|---|---|
| Tufekci et al. 2004 [25] | Premolar bracket (Minnesota Integrated System, American Orthodontics, Sheboygan, WI, USA) | Transbond XT orthodontic adhesive (3M Unitek, Monrovia, CA, USA) | Unspecified generic debonding pliers | Groups (1,2)a—Tungsten carbide bur mounted on a slow-speed handpiece, groups (1,2)b—medium and fine Sof-Lex disks | 20,000 points digitized images obtained by a series of linear profiles (null-point contact stylus system) | Baseline, after debonding and clean-up procedures |
| Banerjee et al. 2008 [28] | Metal orthodontic brackets (3M Unitek, Monrovia, CA, USA) | Non-self-etch resin-adhesive system (Unite, 3M, Unitek, Monrovia, CA, USA) | Unspecified generic debonding pliers | Group 1—slow-speed, eight bladed TC bur (UnoDent, Witam, Essex, UK); Group 2—27 μm of A1 abrasive in an Abradent air-abrasion unit (Crystal Mark, Clendale, CA, USA), 60 p.s.i. o fair pressure, the powder flow was set to 2.2 g/min, with a full power reservoire; Group 3—Abradent unit (same as group 2) and 45S5 bioactive glass (NovaMIne Technology, Alachua, FL, USA) 27 μm < sieved fraction < 53 μm. Until the enamel surface was deemed tob e adhesive-free to visual and tactile examination under ×2.6 magnification (Orascopic HiRes, Sybron Dental Specialities, Orange, CA, USA) | STL files created (TRACECUT24A software) from multiple sections obtained by a contact profilometer (Triclone, Renishaw, Wotton-under-edge, UK), equipped with a 500 μm diameter ruby sphere-tipped stylus (A-5000-7632 KV/ HH, Renishaw, Wotton-under-edge, UK) | Baseline, after debonding and clean-up procedures, after final surface polishing (only the first two were used for volume loss determination) |

**Table 3.** *Cont.*

| | Type of Bracket | Adhesive System | Debonding Procedure | Clean-Up Procedure | 3D Surface Acquisition Device | Acquisition Time |
|---|---|---|---|---|---|---|
| Ryf et al. 2012 [5] | Second molar brackets, 0.022 inch slots (Forestadent, Pforzheim, Germany) | Transbond XT orthodontic adhesive (3M Unitek, Monrovia, CA, USA) | Weingart pliers (3M Unitek, Monrovia, CA, USA) | Group 1—Carbide finishing bur (Maillefer, Ecublens, Switzerland). Group 2—Carbide finishing bur (Maillefer, Ecublens, Switzerland) followed byBrownie Silicone Polisher (Shofu, Kyoto, Japan) and Greenie Silicone Polisher (Shofu, Kyoto, Japan). Group 3—Carbide finishing bur (Maillefer, Ecublens, Switzerland) followed by Astropol F, P and HP polishers (Ivoclar Vivadent AG, Schaan, Liechtenstein); Group 4—Carbide finishing bur (Maillefer, Ecublens, Switzerland) followed by the Renew System Points (Reliance Orthodontic Products, Itasca, IL, USA). Group 5—Carbide finishing bur (Maillefer, Ecublens, Switzerland) followed by Brownie Silicone Polisher (Shofu, Kyoto, Japan) and Greenie Silicone Polisher (Shofu, Kyoto, Japan) and finished with a PoGo polisher (Dentsply, Milford, IL, USA). | 3D imaging device (Laserscan 3D Pro, Willytec GmbH, Grafelfingen, Germany) | Baseline, after debonding and clean-up procedures |
| Janiszewska-Olszowska et al. 2014 [31] | Molar tubes (ERA, Farfield, CT, USA) | Chemical-cure orthodontic adhesive (Unite, 3M, USA) | Ligature cutting pliers | None | 3D optical scanner (Atos III, Triple Scan, GOM, Germany) | Baseline and after debonding procedures |

**Table 3.** *Cont.*

| | Type of Bracket | Adhesive System | Debonding Procedure | Clean-Up Procedure | 3D Surface Acquisition Device | Acquisition Time |
|---|---|---|---|---|---|---|
| Suliman et al. 2015 [29] | Group 1—metal reinforced polycristalline ceramic brackets (Clarity, 3M Unitek, Monrovia, CA, USA); Group 2—clear monocrystalline ceramic brackets (Inspire-ICE, Ormco, Orange, CA, USA) | Transbond XT orthodontic adhesive (3M Unitek, Monrovia, CA, USA) | Group 1—Weingart pliers (OrthoPli, Philadelphia, Penn); Group 2—recommended plastic debonding instrument (Omrco, Orange, CA, USA) | High-speed handpiece and multi-fluted carbide bur (H48LQ, Komet of America, Schaumburg, III) | 3D optical scanner (COMET xS, Steinbichler Vision System, Neubeuern, Germany) | Baseline, after debonding and clean-up procedures |
| Janiszewska-Olszowska et al. 2015 [32] | Molar tubes (ERA, Farfield, CT, USA) | Chemical-cure orthodontic adhesive (Unite, 3M, USA) | Ligature cutting pliers | Group 1—twelve-fluted tungsten carbide bur (123-603-00, Dentaurum, Pforzheim, Germany), Group 2—one-step finisher and polisher (inverted cone One gloss, Shofu Dental, Kyoto, Japan), Group 3—adhesive residue remover (989-342-60, Dentaurum, Pforzheim, Germany) | 3D optical scanner (Atos III, Triple Scan, GOM, Germany) | Baseline, after debonding and clean-up procedures |
| Stadler et al. 2019 [33] | Conventional bracket (Victory Series, 3M, St.Paul, MN, USA) | Opal bond (Ultradent, South Jordan, UT, USA) | Bracket removing plier (678-220L, Hu-Friedy, Chicago, IL, USA) | Six-blade tungsten carbide bur (H23RA, Gebr. Brasseler GmbH, Lemgo, Germany) mounted in a low-speed contra-angle handpiece (KaVo Master Series, Biberach, Germany), multistep Sof-Lex discs (coarse, medium, fine, super fine, Sof-lex, 3M Unitek, Monrovia, CA, USA) | 3D optical scanner (Cerec Omnicam, Software SW 4.5,1 Dentsply Sirona, York, PA, USA) | Baseline, after debonding and clean-up procedures |
| Engeler et al. 2022 [35] | Buccal tooth with conventional brackets (Victory Series, 3M, St. Paul, MN, USA) and lingual tooth with customized brackets (Incognito™, 3M, St. Paul, MN, USA) | Transbond XT orthodontic adhesive (3M Unitek, Monrovia, CA, USA), Opal bond (Ultradent, South Jordan, UT, USA), Bracepaste (American Orthodontics, Sheboygan, WI, USA) | Not specified | Tungsten carbide bur (H23RA, Gebr. Brasseler GmbH, Nord-Rhine Westpahlia, Germany) mounted on a low-speed handpiece (KaVo Master Series, Baden Württenberg, Germany) first with water cooling and then with air cooling. | 3D surface scanner (inEos X5, Dentsply Sirona, York, PA, USA) | Baseline and after clean-up procedures |

**Table 3.** *Cont.*

| | Type of Bracket | Adhesive System | Debonding Procedure | Clean-Up Procedure | 3D Surface Acquisition Device | Acquisition Time |
|---|---|---|---|---|---|---|
| Cesur et al. 2022 [34] | Group 1—metal brackets (Ormco Mini Diamond Twin brackets, Ormco, Orange, CA, USA), Group 2—ceramic brackets (Inspire-ICE, Ormco, Orange, CA, USA) | Transbond XT primer (3M Unitek, Monrovia, CA, USA) Transbond XT orthodontic adhesive (3M Unitek, Monrovia, CA, USA) | Bracket-removing pliers (Hu-Friedy, Chicago, IL, USA) | Group A—8-blade tungsten carbide burs (Komet, Lemgo, Germany) at low speed (10,000 rpm) until visually clean. Polishing was performed with rubber cups (Nais, Sofia, Bulgaria) and pumice at 5000 rpm for 30 s. Group B—Fiber-reinforced stainbuster composite burs (Abrasive Technology Inc., Lewis Center, OH, USA) (10,000 rpm) untill visually clean. 30 s of polishing at 5000 rpm was performed using rubber cups with Detartrine paste (Septodent, France). Group C—Coarse (10,000 rpm, as needed), fine (10,000 rpm, 15 s), and ultrafine (30,000 rpm, 15 s) Sof-Lex discs (3M Dental, St Paul, MN, USA) | 3D reconstruction from Micro-CT scans | Baseline, after debonding and clean-up procedures |

**Table 4.** Included studies and assessed variables, part 2.

| | Type of Model | Digital Analysis Software | Type of Specimens | Minimum Mean Volumetric Loss (mm$^3$) per Tooth | Group (min MVL) | Maximum Mean Volumetric Loss (mm$^3$) per Tooth | Group (max MVL) |
|---|---|---|---|---|---|---|---|
| Tufekci et al. 2004 [25] | 3D reconstruction of harmonious profiles | AnSur NT software (Regents, University of Minnesota, MN, USA) | 28 extracted human premolars, 14 with artificially created white spot lesions and 14 without it. All specimens were placed with green die stone (Die Keen, Modern Materials/Heraus Kulzer, Armonk, NY, USA) in nylon rings, while the labial third of the crown and the cervical portion of the root were indicated for the bonding and debonding procedures | 0.06 (SD = 0.04) | Group 1A | 0.17 (SD = 0.103) | Group 1B |



**Table 4.** *Cont.*

| | Type of Model | Digital Analysis Software | Type of Specimens | Minimum Mean Volumetric Loss (mm³) per Tooth | Group (min MVL) | Maximum Mean Volumetric Loss (mm³) per Tooth | Group (max MVL) |
|---|---|---|---|---|---|---|---|
| Banerjee et al. 2008 [28] | Resin replicas of the buccal surfaces (Araldite 2015, Huntsman Advanced Materials, Evenberg, Europe) | Geomagic studio 8 (Geomagic, NC, USA) | 30 extracted intact human premolars with sound surfaces, sectioned horizontally, 2 mm below the CEJ and located on a Perspex block through thermoplastic compound (Tecbond, Kenyon group, Lancashire, UK) leaving bared the buccal surface | 0.135 (SD = 0.033) | Group 3 | 0.386 (SD = 0.254) | Group 2 |
| Ryf et al. 2012 [5] | Dental stone cast (Fuji Super Hardrock, GC, Leuven, Belgium) and 3D mode | Match-3D software (StemmerImaging, Puchheim, Germany) | 75 extracted human molars | 0.19 (SD = 0.15) | Group 3 | 0.26 (SD = 0.15) | Group 4 |
| Janiszewska-Olszowska et al. 2014 [31] | 3D scan | GOM Inspect software (GOM, Braunschweig, Germany) | 15 third molars without carious lesions, extracted for orthodontic reasons from 16–24 years patients. To avoid useless movement, the human teeth were fitted in impression silicone (Bisico S1 Soft, Bisico, Germany) | None | None | None | None |
| Suliman et al. 2015 [29] | 3D scan | Cumulus software (Regents of the University of Minnesota, MN, USA) | 40 extracted intact human premolars | 0.238 (SD = 0.136) | Group 2 | 0.420 (SD = 0.287) | Group 1 |
| Janiszewska-Olszowska et al. 2015 [32] | 3D Scan | GOM Inspect software (GOM, Braunschweig, Germany) | 30 third molars without carious lesions, extracted for orthodontic reasons from 16–24 years human patients. To avoid useless movement, the human teeth were merged into in impression silicone (Bisico S1 Soft, Bisico, Germany) | Not reported | Group 3 | Not reported | Group 1 |

**Table 4.** *Cont.*

| | Type of Model | Digital Analysis Software | Type of Specimens | Minimum Mean Volumetric Loss (mm³) per Tooth | Group (min MVL) | Maximum Mean Volumetric Loss (mm³) per Tooth | Group (max MVL) |
|---|---|---|---|---|---|---|---|
| Stadler et al. 2019 [33] | 3D scan | OraCheck (Version 2.13.8676, Cyfex AG, Zurich, Switzerland) | 120 extracted permanent intact bovine incisors. 12 upper dental arches were produced fitting 10 teeth (from tooth 15 to 25) on a wax plate, with interproximal contacts as similar as possible to the maxillary dental arch, stuck through hot-setting glue, and merged into hot polymer (ProBase, Ivoclar Vivadent AG, Shaan, Liechtestein) | 0.17 (SD = 0.21) | Group 2A | 0.61 (SD = 0.51 (1B), SD = 0.37 (2B)) | Groups 1B and 2B |
| Engeler et al. 2022 [35] | 3D tooth model | OraCheck software (Version 2.13.8676, Cyfex AG, Zurich, Switzerland) | 56 extracted human permanent teeth were collected. Two maxillary and two mandibular dental arches were developed: Fourteen teeth ranging from 17 to 27 and 37 to 47 respectively were positioned in their intra-arch locations with interproximal contacts as similar as possible to a dental arch in a wax plate. They were stuck through hot-setting-glue, and merged into a hot polymer base (ProBase, Ivoclar Vivadent AG, Schaan, Liechtenstein). A gingiva wax mask was molded (BELLADI Superior Rosa, Belladi Ruscher Schleusser, Amriswil, Switzerland) and then replaced with a silicone material (Finogum Premium, Fino, Bad Bocklet, Germany) | 0.34 | NON-FIT | 0.56 | FIT (BRACE) |
| Cesur et al. 2022 [34] | Micro-CT 3D reconstruction | CTAn (SkyScan, Bruker, Billerica, MA, USA) | 42 extracted maxillary first premolars with no visible fractures, caries or restoration. The crown and rooths were carefully separed | 0.02 (SD = 0.01 (1B), SD = 0.00 (1C)) | Groups 1B and 1C | 0.11 (SD = 0.18) | Group 2A |

All investigations were published between 2004 and 2022. No in vivo papers were retrieved. No papers investigating clear aligners attachment clean-up were retrieved. Four researches were conducted on human premolars [25,28,29,34]. Three studies used human molars [5,31,32], one analyzed all types of permanent teeth [35], and, in one investigation, bovine permanent incisors were selected as specimens [33]. Metal and ceramic brackets were distributed among included studies as follows: metal brackets were chosen in seven studies [5,25,28,31–33]; ceramic brackets were used in one; and both metal and ceramic brackets were selected in one paper. Only one paper investigated lingual appliances [35]. For the debonding procedure, all studies removed brackets conventionally with the use of a different hand. The clean-up procedure varied among all investigations. All studies that performed the clean-up phase included at least one group with multi-fluted tungsten carbide burs, making it the most investigated tool.

In order to assess the volume loss related to debonding and clean-up procedures, all studies performed a volumetric acquisition at baseline and one at the end of all procedures. Six articles used an optical scan [5,29,31–33,35], one used a null-point stylus system [25], one used a Micro-CT scan followed by a 3D file reconstruction [34], and one used a contact profilometer [28]. Both baseline and after debonding/clean-up acquisitions were superimposed, and the volumes were subtracted using different digital analysis software. Among all included studies, the volumetric loss of enamel ranged from 0.02 (SD = 0.01) mm$^3$ to 0.61 (SD = 0.51) mm$^3$ per tooth.

## 4. Discussion

Debonding procedures continue to represent a debate topic among dental practitioners. There is not consensus over which tools and protocols should be preferred to detach brackets and remove cement residues [3]. All of the techniques chosen by the authors of the included papers were shown to cause enamel damage of different degrees. This review focuses on the volumetric analysis of those enamel defects.

### 4.1. Enamel Loss Following Debracketing

Among the included papers, enamel damage has been reported by some authors to be caused both by debonding and clean-up procedures [5,29,31,34]. Janiszewska-Olszowska et al. analyzed the enamel damage due to bracket removal and reported the bond failure at the interface between enamel and adhesive produces enamel loss [31]. Bond failure may occur: (1) between the bracket's surface and the adhesive; (2) within the composite; and (3) between the composite and the enamel. The last one may result in enamel substance loss of a certain amount [15] and could require direct restorations to be applied to solve the esthetic issue [37]. The enamel damage area and the volume of surface loss also depends on the size of the orthodontic bracket and on the type of material [6,29,34]. Using small brackets could be less detrimental since a lower force will be needed to detach them [6]. The bracket's material is also important, as reported by Suliman et al. [29] They compared, in fact, metal reinforced polycrystalline and clear monocrystalline ceramic brackets. The authors showed that most of the polycrystalline brackets fractured in two or more pieces when detached. The fragments remaining bonded to the tooth may cause the clean-up procedure to be more invasive, time consuming, and cumbersome. The clear monocrystalline brackets, on the other hand, resulted to be more resistant to fractures. This finding was attributed to the zirconia microspheres embedded in the bracket's base [29]. In the same study, two types of pliers were tested: the conventional Weingart pliers for the polycrystalline group and the recommended plastic debonding instrument for the monocrystalline group. The manufacturers claim that the latter allows to peel the bracket off from the tooth in one piece. This may have contributed to experiencing fewer fractures in this group [29]. In the first group, however, the use of Weingart pliers may have represented one cause of higher fracture frequency [9].

Cesur et al. compared monocrystalline ceramic brackets with conventional metal brackets [34]. They found that a significantly greater demineralization volume occurred

in the ceramic bracket group. Ceramic bracket debonding and consequent damage to enamel surfaces has been investigated over the years for orthodontic research. Moreover, many studies have demonstrated the negative effects of ceramic brackets on the enamel surface, supporting Cesur's findings [9,29,38,39]. Ryf et al. found that out of a mean enamel defect (post clean-up) of $+/- 0.22$ mm$^3$, 0.02 mm$^3$ were lost in the debracketing phase [5]. Moreover, they reported a volume loss of 0.13 mm$^3$ in a sample with ARI score 1, a mean volume loss of 0.02 mm$^3$ when the ARI score was 2, and no volume loss when ARI score was 3, suggesting a correlation between the adhesive remnants and the enamel loss. Nevertheless, these findings must be considered carefully since this paper has been assessed with a high risk of bias.

### 4.2. Enamel Loss Following Clean-Up Procedures

Even though with a low ARI score the debonding may have a relevant influence, adhesive removal (clean-up) is the phase that produces the highest amount of enamel damage [29]. There are many tools available on the market for the adhesive removal and, although there is no agreement on which protocol to apply in this stage, it is believed that multi-step cleaning produces less damage than single-step procedures [19,40]. Ryf et al. concluded that clean-up procedures with carbide burs alone may cause excessive enamel loss and leave a large amount of composite on the surface. Conversely, when combined to the multi-step rubber polishing kits, it showed some advantages in enamel loss prevention. Even though composite remnants were not completely removed, it gave a smooth and shiny surface that was visually assessed as an accurate clean-up [5]. Banerjee et al. compared the efficacy of tungsten carbide burs with air abrasion [28]. They reported that air-abrasion with alumina particles was shown to cause more damage than the tungsten carbide bur, and the amount of enamel removed was far less predictable. Based on this finding, they stated that alumina particles are not suited for debonding procedures. Conversely the bioactive glass air-abrasion was found to be at least as good as the tungsten carbide bur in terms of enamel preservation [28]. The inherent characteristics and the clinical technique are also important. The divergent stream cuts mainly at the center and less at the periphery, resulting in an indistinct abrasion margin, making it easier to polish. In order to obtain those properties a proper distance and technique need to be used [28]. In the attempt to find the best tool, which should be able to be selective for the adhesive without harming the sound enamel, Cesur et al. tested a zircon-rich glass-fiber-reinforced composite bur [34]. This bur was developed for other purposes, but various studies showed that it can be safely used for debonding [17,41]. They concluded that this composite bur is less detrimental than the tungsten carbide bur. Moreover, after debonding of conventional metal orthodontic appliances, composite bur followed by either polishing pastes or Sof-lex discs offered the best protection for enamel. Conversely, after ceramic bracket debonding, the authors reported that Sof-lex discs resulted as the most appropriate resin removal method [32]. Although lingual appliances represent a fixed alternative to clear aligners, especially for complex cases [42], only one paper investigated them [35]. The authors, focusing on clean-up aids, did not report if buccal or lingual brackets resulted in significantly higher enamel loss.

In order to detect and remove adhesive properly, Stadler et al. and Engeler et al. tested a fluorescence-aided identification technique (FIT), comparing it with a conventional light source [33,35]. Stadler et al. reported a high sensitivity of the FIT method which allows to reveal small composite remnants in difficult to find areas, for example, in grooves or pits. This technique also tolerates water cooling, thus limiting a potential temperature increase that may harm the pulp during a dry clean-up [43]. This finding was also supported by Engeler et al., who reported higher clean-up efficacy when FIT was used. Conversely, lingually, the FIT methods resulted in larger enamel defects. Stadler et al. also reported that the FIT method reduced the operating time significantly. The time required per tooth was on average 80 s with the FIT technique, whereas it was 130 s approximately with conventional light source [33].

Among the included studies, the enamel volume loss following debonding ranged between $0.02 \pm 0.01$ mm$^3$ [34] and $0.61 \pm 0.51$ mm$^3$ [33]. Those results have different clinical significance based on the considered tooth. The same volume lost on a molar surface may have different significance compared to lower incisors.

*4.3. Limitations of This Study*

A limitation of this study is the small amount of included papers. In order to consider the volume of enamel defects, many articles considering only the linear depth of enamel defects were excluded. Another limitation of the present study is related to the medium/high risk of bias of the included papers, which precludes robust conclusions.

**5. Conclusions**

Based on the findings of this systematic review, the following conclusions can be drawn:

1.  The volumetric loss of enamel after debonding and clean-up procedures ranges from $0.02 \pm 0.01$ mm$^3$ to $0.610.51$ mm$^3$ per tooth.
2.  The debonding/clean-up procedures which are able to cause the least enamel volume loss are still controversial.

**Author Contributions:** Conceptualization, G.P. and M.M.; methodology, S.B.; software, M.M.; validation, G.C., E.G., and G.G.; formal analysis, G.P.; investigation, M.M.; resources, S.B.; data curation, M.M. and S.B..; writing—original draft preparation, G.P. and M.M.; writing—review and editing, M.G.P.; visualization, G.C., C.G. and S.S.; supervision, E.G. All authors have read and agreed to the published version of the manuscript.

**Funding:** This research received no external funding.

**Institutional Review Board Statement:** Not Applicable.

**Informed Consent Statement:** Not applicable.

**Data Availability Statement:** Not Applicable.

**Conflicts of Interest:** The authors declare no conflict of interest.

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
