# Peer review of "Quantitative Volumetric Enamel Loss after Orthodontic Debracketing/Debonding and Clean-Up Procedures: A Systematic Review"

_applsci, doi:10.3390/app13095369_

Round 1

Reviewer 1 Report

Thank you for the invitation to review the manuscript entitled "Quantitative volumetric enamel loss after orthodontic de bracketing/debonding and clean-up procedures. A systematic Review."

The manuscript is a comprehensive review of 4 databases, investigating the amount of volumetric loss that results from debracketing and debunking procedures at the end of an orthodontic treatment.

The study is well written, well organized and well conducted.

English translation of certain sentences in the introduction is not that smooth and understandable. 

The manuscript is a systematic review assessing the volumetric enamel loss secondary to debracket and debonding procedures after orthodontic treatment. It is well conducted and it correctly follows all the methodological requirements of a systematic review. The topic investigated by this systematic review is very important, as enamel loss is likely to increase the risk of cavities and weakness of the dental structure. They also pointed out a specific gap in the literature, in that they were not able to find any studies addressing the same topic after invisalign treatment. This is also very interesting. Publishing this paper would be important as it identifies current gap in the available literature as well as it brings attention to the damage of debonding procedures after routine orthodontic treatment. Conclusions are consistent with the results; references are adequate in number and in content. I do not request any further improvement from the authors.

Author Response

The author thanks the reviewer for the time spent in reviewing the manuscript.

English has been improved as suggested.

Reviewer 2 Report

The study is well designed and brings relevant information for the clinicians.

Author Response

The author thanks the reviewer for the time spent in reviewing the manuscript.

Reviewer 3 Report

The manuscript's introduction clearly states the problem and why it is important. The methodology is well structured, and the results are presented clearly. The discussion is appropriate and in accordance with the results. Please check the grammar and writing. Some phases of phrases could be easier to read. Nice work!

Author Response

The author thanks the reviewer for the time spent in reviewing the manuscript.

Some sentences have been improved as suggested.